# On the Collaborative Use of EV Charging Infrastructures in the Context of Commercial Real Estate

**Joela Gauss †, Sascha Gohlke †, Zoltán Nochta \*,†**

Data-Centric Software Systems (DSS) Research Group, Institute of Applied Research (IAF),
Karlsruhe University of Applied Sciences, 76133 Karlsruhe, Germany
\* Correspondence: zoltan.nochta@h-ka.de
† These authors contributed equally to this work.

**Abstract:** Resource sharing in general is a means of solving the problem of infrequent and, thus, inefficient utilization of expensive or scarce resources. In this paper, we present an approach to run shared EV-charging infrastructures in the context of commercial real-estate facilities. Collaborating EV-charger owners thereby create a pool of chargers for shared use. In our work, we consider aspects of economic viability, desirability and technical feasibility as prerequisites of a successful solution. We formally prove that the basic economic potential of the proposed pooling with regard to overall infrastructure utilization is given. In order to operate the shared pool of charging points at a given location, the corresponding management software must fulfil specific requirements. Our prototype implementation that was realized as an extension of the open-source system Open E-Mobility demonstrates the technical feasibility of the sharing idea in a user-friendly way. Questionnaires and personal interviews conducted with owners of small and medium-sized businesses revealed that they would share charging stations if it helped overcome availability bottlenecks, thus improving customer and employee perception.

**Keywords:** EV charging infrastructure; pooling; sharing; management software; processes





## 1. Introduction

Thanks to huge investments made by the automotive industry as well as the political and financial support given by government incentives worldwide, the number of electric vehicles (EV) will continue to rise in the coming years. Therefore, the rapid creation of sufficient EV charging capacity for diverse mobility scenarios is required. For instance, in Germany, by the end of 2021, there were ca. 52,000 publicly accessible charging points (CP) registered [1] serving estimated one million EVs (PHEV and BEV) in the country [2]. The German government's goal is to reach up to 15 million EVs by 2030 [3]. Currently, the installation and operation of *public* EV charging stations—especially in urban areas— appear to be economically unattractive, with the result that the required upscaling effects have so far failed to materialize. It is similarly problematic to scale the number of *private* charging points ("wall boxes"), which are mostly installed in single-family homes in the suburbs: If large numbers of EVs were to be charged at the homes on the same street simultaneously, e.g., during the after-work hours, the local power supply could reach its limits. As a countermeasure, local grid operators can increase grid connection capacity by installing additional cabling, transformers, or by implementing means of remote control to actively manage EV charging. Both options are highly cost intensive as well as having low profit margins. Due to these limitations, in the near future, *semi-public* parking areas and charging stations at office buildings, factories, industrial areas, shopping centers, hotels, etc., might play a significant role in "fuelling" the growing EV population. Such charging facilities primarily power EVs, which are used for business purposes of a company, including, for example, service cars, delivery vans, and shuttles.

At the same time, the charging stations can also cover the electricity demand of privately used vehicles to a huge extent, for example, EVs of employees, customers, visitors, or even neighbours if they are authorized to enter the respective parking area. Thus, companies' semi-public charging infrastructures can help relieve the entire (public) power system [4] and also reduce required investments in public charging infrastructures.

Commercial real-estate facilities and corresponding in- or outdoor parking areas are often used by several tenants simultaneously. For example, the first floor of an office building might house a medical practice and a restaurant, while the upper floors contain offices rented by different corporations. Usually, each tenant has dedicated, reserved car parking lots for its employees, customers, business partners, and visitors. As EVs become increasingly attractive, tenants may want to establish exclusively used EV chargers on their own parking spots. In real-world scenarios, however, the exclusive assignment of EV-charging equipment to single tenants can be challenging due to both economic (high TCO per charging station, low utilization outside of business hours) and technical reasons (insufficient electrical infrastructure in the building, power limitations) [5–9]. The collaborative use of charging stations that are shared among multiple independent tenants of a building might help overcome these problems.

In our approach, collaborating CP owners make their own installed equipment basically available and accessible to each other. The resulting *pool of charging points* helps cover peak demand time intervals of all participating contributors, and it also helps increase the overall utilization of the entire charging infrastructure.

For a better understanding, Figure 1 shows an example scenario for the proposed pooling of CPs in the context of a building with an integrated parking space in the basement. Due to technical limitations, the building can host a maximum of 15 charging points. With e-mobility in mind, the building owner has initially foreseen 15 cable trays for future EV charger installations and installed four charging points before opening the building. Since then, CP 1 has been jointly used by a physician (on weekdays during the daytime) and by a restaurant (in the evening hours and on weekends), while CPs 2, 3 and 4 are rented by a company termed as "Tenant 1".

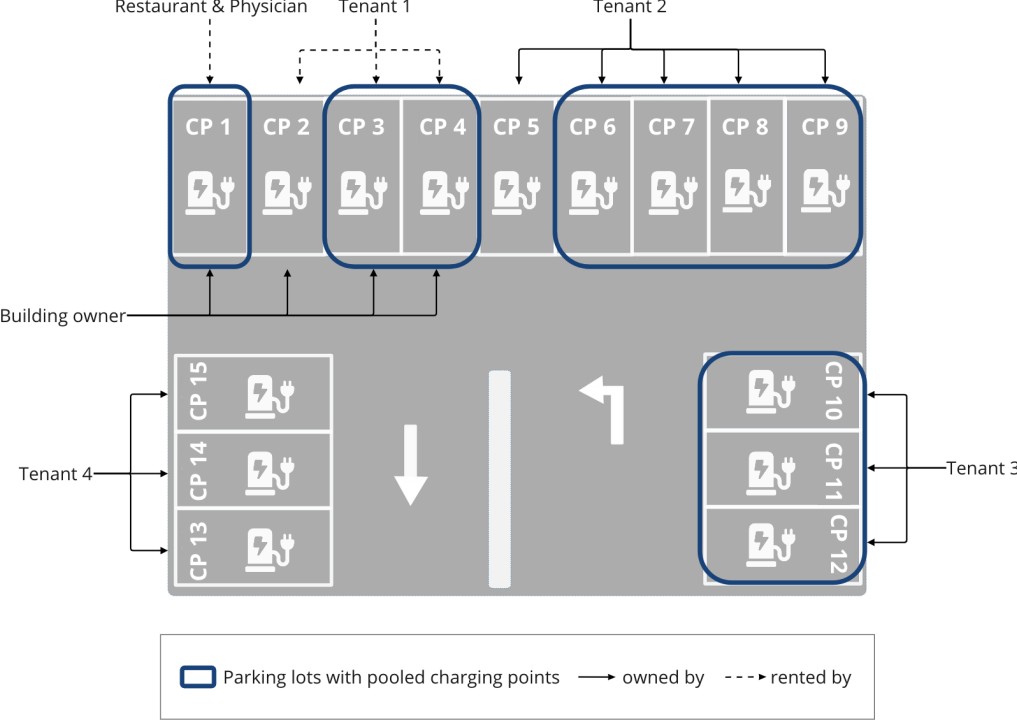

**Figure 1.** Example CP pooling scenario in an office building's parking area.

Over time, Tenants 2, 3 and 4 had been equipping their parking slots with their own EV chargers because the building owner himself did not want to invest in additional equipment. Meanwhile, the available charging points of single tenants are not sufficient anymore to cover their respective demand. They are faced with availability problems especially at peak times specific to their businesses. Employees, customers, guests and visitors often cannot find an available charging point at the given company's dedicated parking lots, while at the same time chargers of other tenants are unused. To improve the frustrating situation, some tenants joined forces and created a shared pool of charging points. The pool consists of 10 (out of 15 installed) charging points because Tenant 1 and Tenant 2 only partially share their CPs, and Tenant 4 does not participate in the CP pooling initiative at all. The CP pool and related processes are managed with the help of a software system that enables independent participants to individually configure their specific settings and preferences.

This paper is structured as follows: In Section 2, we revisit publications that deal with resource sharing and problems of EV charging. Section 3 summarizes the methodology we have applied in our research project. In Section 4, we explain the benefits of CP pooling and provide a mathematical proof of why sharing partners can expect higher utilization of their charging points. Section 5 introduces main processes, describes identified system requirements, and provides details about the prototype implementation of a software system, which enables the management of pooled CPs at a given location. Section 6 presents the results of a two-part end-user survey, which we conducted with small and medium-sized businesses in a small German town. Finally, in Section 7, conclusions are drawn, and future work is outlined.

## 2. Related Work

In the area of electric vehicle charging, most of the recent research has focused on solving problems related to power grid constraints and the usage of power at affordable cost. The proposed solutions often combine energy consumption and distribution strategies (e.g., load shifting, peak shaving) with demand response management techniques (e.g., dynamic time-of-use tariffs) to reduce peak demand and optimize the total cost of energy used in enterprise fleets [10–14]. Another approach to reduce the costs is to save energy in a fleet by applying platooning under optimized parameters, such as inter-vehicle gaps [15]. Dynamic pricing systems and related load shifting solutions can help avoid critical electricity demand peaks, but they are not suitable to improve the situation regarding the basic availability of charging points at a specific location. Regarding the issue of availability, previous research has examined the problem of parking spaces and charging stations being blocked by vehicles that are not charging. In [16], a time-based usage fee of charging stations is proposed providing an incentive to drive away after the charging was completed. To automatically detect (unplugged) vehicles that block free parking spaces, the use of proximity sensors is suggested in [17]. Our research and the proposed approach target this gap and help improve the overall availability of charging points through sharing.

In conjunction with the shared use of material goods and immaterial services, the terms *sharing economy* and *collaborative consumption* are widely used in academic literature. According to [18], "Sharing economy consists of the practice of use and share of products and services with two or more individuals, with or without the transfer of ownership, with no material compensation (neither non-monetary compensation) and mediated through social mechanisms". Our CP pooling approach partly fits this definition because pooled charging points are made accessible to other pooling partners without transferring ownership. There is a difference with regard to the pooling agreement, which is negotiated between the participants to regulate the co-usage under fair conditions upfront, including the determination of usage fees as means of compensation.

Usage fees are seen in [18] as a property of collaborative consumption, i.e., "transactions where people coordinate the exchange of goods and services for a fee or other compensation (monetary or non-monetary), where a triadic is existing among a platform provider, peer service provider and a customer, there is no ownership transfer and it is mediated through market mechanisms". However, in our case, the jointly negotiated pooling agreement eliminates the need for involving a platform provider as a mediator. Due to the predetermined fees for using pooled CPs, a mediation through market mechanisms is excluded as well. The term "sharing" is also frequently used in context of ad-hoc and short-term rental services, such as sharing cars or e-scooters in cities [19]. In [20], Belk unmasks and criticizes those rather commercial activities as "pseudo-sharing". In this spirit, we consider our pooling approach as "real sharing" because all involved participants can both provision and consume shared resources under previously negotiated (and therefore fair) conditions. The sharing of expensive and less frequently used machinery is common practice in the agricultural domain [21]. A simple manifestation of it is "mutual aid" between neighboring farmers who help each other out with equipment. Thus, so-called "machinery rings" are established to coordinate the demand-driven cross usage of machinery, for example, to temporarily replace a broken machine. Members of "machinery partnerships" jointly purchase equipment and define operational policies. Our approach can be seen as a mixture of these variants, as neighbouring CP owners help each other out especially to cover EV-charging demand at peak usage hours, while they buy and manage their own equipment without involving a coordinating party.

The concept of allowing others to access privately installed charging points is found in existing products and also in the academic literature. As an example, the Webasto ChargeConnect App [22] allows the owner of a private charging point to grant a time-limited access to guest users. The guest must first register in the system and be authorized by the owner of the CP in question to start the charging process. The use of a third-party platform to coordinate the sharing of private charging points is also discussed in [23,24]. With the help of the demonstrator "CrowdStrom", CP owners provide access to other EV drivers in return for a financial compensation. The price for using a charging point is set on an hourly basis, and authorized EV drivers can reserve advertised charging points before they arrive. This approach can also be characterized as a short-time rental service rather than "real" sharing of charging points among equal peers.

According to survey results, 30–40% of BEV owners use charging facilities at their workplaces, of which 8% exclusively charge at work [25]. In the context of semi-public infrastructures, the development of an operational model, which involves real estate agencies in the installation and operation of charging infrastructures, was considered in [26]. Simulations estimate a significant increase of charging power demand of EVs, based on data captured at shopping centers [27]. Peak electricity demand can be reduced, for instance, by applying dynamic pricing, which makes charging at "peak-times" increasingly unattractive for EV drivers [7]. A broad application of pooling and the resulting increased utilization of charging points could enable basically more drivers to charge their EVs at work. Charging more EVs during the business hours at work can help reduce the power demand at homes, especially during the after-work hours, as advised by Alatise et al. in [4]. The importance of reducing (or shifting) charging demand at home is also underpinned by Gaikwad, who simulated daily charging profiles and predicted substantial transformer overloading, when most cars are charged at the same time [28].

## 3. Methodology

Our basic working hypothesis is that the shared usage of charging points will only be adopted by the market in the future if the *economic viability*, *technical feasibility* and *desirability* of the concept and its practical implementation are given [29]. Accordingly, we dealt with all three aspects in parallel in our project:

- **Economic viability:** We created an abstract definition of resource sharing and formally proved the hypothesis that the shared usage of charging points will most likely

improve the overall utilization of charging infrastructures. As a consequence, CP pooling helps make related investments more profitable for all involved CP owners.

- **Technical feasibility:** We collected and analysed technical requirements along with non-functional features that charge point operating systems should provide to serve a shared pool of CPs. To prove the technical feasibility of CP pooling, we have decided to implement related requirements as extensions of the open source software system "Open E-Mobility" [30], which is used to operate charging infrastructures in several real-world deployments. Adding new functionality to an existing software system (also known as the brownfield approach) instead of building a dedicated solution (greenfield development) not only helps reduce development costs, but also allows customers to move from exclusively used CPs to shared scenarios in future without changing their existing system. The resulting extended software system was deployed and tested in a prototype environment. The tests covered technical aspects mainly related to the functional correctness of the software system. The implemented prototype system has also been used in five qualitative face-to-face interviews, which we carried out with four retailers and a restaurant owner to gather early feedback from potential end-users.

- **Desirability:** We have collected preferences of potential end-users related to the shared usage of charging points. For this purpose we initially conducted personal interviews and questionnaires with selected business owners in a small German town, consisting of retailers (~30%), service providers (~24%), manufacturers (~10%) and others. The questionnaire was mainly used to find out whether and under which conditions companies would allow others (incl. customers, visitors, employees of other companies, etc.) to access their EV charging equipment.

## 4. Economic Viability of Sharing Charging Points

The proposed pooling is an approach to bundle charging points that are owned and operated by multiple independent entities (typically businesses). In general, a pool of charging points is not limited to a specific geographic location or context, such as the parking area of a single office building. Instead, a pool may include charging points that are installed at multiple locations of the pooling participants, such as larger companies that usually have offices in different cities or even countries. Establishing a pool increases the basic availability of charging points to individual EV-drivers, e.g., employees, guests and customers of pooling participants, as they gain access to all pooled charging points at their respective location. From a driver's perspective, the creation of a pool of CPs leads to a higher likelihood to find an available charging point at the given location basically at any point of time. In addition to the obvious benefits for EV drivers, the pooling of charging points can also increase the utilization of the overall charging infrastructure. If a particular charging point is exclusively used by only one business, it could remain completely unused outside of that company's regular business hours, i.e., up to 120 h per week. There is also no guarantee that such a bound charging point will be used continuously during the owner's business hours, for example, when the EV-driving employees are on vacation.

The increased utilization of pooled charging points compared to exclusively used equipment can be formally proven as follows:

Let $M$ with ($|M| \in \mathbb{N}$) the set of installed charging points within a charging infrastructure and $I$ with ($|I| \in \mathbb{N}$) the set of CP owners (who contribute to the pool).

$M_i$ denotes the set of charging points of a given owner, whereas $M = \cup_{i \in I} M_i$ ($M_i \cap M_j = \varnothing, i \neq j$) holds for the entire infrastructure at the facility, meaning that each CP has only one owner.

Let $N_i$ be the set of EV drivers, who are assigned to the charging points of a given CP owner. It is assumed that $N = \cup_{i \in I} N_i$ ($N_i \cap N_j = \varnothing, i \neq j$) holds because normally an EV driver is an employee (business partner, customer, visitor) of only one company at the same time. $N_i(t) \subseteq N_i$ is the set of EV drivers that are assigned to a given CP owner and require a charging point at time $t \in \mathbb{R}$. Thus, $N(t) = \cup_{i \in I} N_i(t)$ is the set of all EV drivers that look

for an available charging point at $t$. $M_i(t) \subseteq M_i$ denotes the set of charge points of a given owner that are used at time $t$ and $M(t) = \cup_{i \in I} M_i(t)$ is the set of all used CPs at $t$.

The function $l(t) : \mathbb{R} \to \mathbb{N}$, $l(t) = |M(t)|$ shows the level of utilization, i.e., the total number of charging points that are used at time $t \in \mathbb{R}$.

Thus, $L = \int_{-\infty}^{\infty} l(t) \, dt$ stands for the total utilization of the charging infrastructure over time.

$\widetilde{L}$ stands for the utilization of the charging infrastructure in which all CPs are pooled, while $\underset{\sim}{L}$ represents the same CPs' utilization in case each of them is used exclusively by just one CP owner, i.e., without being added to a pool. Accordingly, $\widetilde{M}(t)$ is the number of actually used CPs over time, which are part of the pool, while $\underset{\sim}{M}(t)$ stands for the "non-pooled" case, i.e., no CP owner shares any of its charging points.

**Theorem 1.**
$$\widetilde{L} \geq \underset{\sim}{L} \tag{1}$$

**Proof.**
$$\widetilde{L} = \int_{-\infty}^{\infty} \widetilde{l}(t) \, dt \tag{2}$$

$$= \int_{-\infty}^{\infty} |\widetilde{M}(t)| \, dt = \int_{-\infty}^{\infty} min\{|M|, |N(t)|\} \, dt \tag{3}$$

$|\widetilde{M}(t)| = min\{|M|, |N(t)|\}$ holds because, when using a CP pool, the number of actually usable charging points at a given time is basically limited by the total number of pooled charging points of all CP owners:

$$= \int_{-\infty}^{\infty} min\{\sum_{i \in I} |M_i|, \sum_{i \in I} |N_i(t)|\} \, dt \tag{4}$$

$|M| = \sum_{i \in I} |M_i|$ because considering disjoint charging points $M = \cup_{i \in I} M_i$ holds.
$|N(t)| = \sum_{i \in I} |N_i(t)|$, since assuming a disjoint driver population $N(t) = \cup_{i \in I} N_i(t)$ holds.

$$\geq \int_{-\infty}^{\infty} \sum_{i \in I} min\{|M_i|, |N_i(t)|\} \, dt \tag{5}$$

$\sum_{i \in I} min\{a_i, b_i\} \leq min\{\sum_{i \in I} a_i, \sum_{i \in I} b_i\}$, is a consequence of $\sum_{i \in I} min\{a_i, b_i\} \leq \sum_{i \in I} a_i$

$$= \int_{-\infty}^{\infty} \sum_{i \in I} |\underset{\sim}{M}_i(t)| \, dt \tag{6}$$

$|\underset{\sim}{M}_i(t)| = min\{|M_i|, |N_i(t)|\}$ holds because, if charging points are not shared, an EV driver can only occupy an available CP of exactly one CP owner (e.g., her employer) that she is assigned to:

$$= \int_{-\infty}^{\infty} |\underset{\sim}{M}(t)| \, dt \tag{7}$$

$|\underset{\sim}{M}(t)| = \sum_{i \in I} |\underset{\sim}{M}_i(t)|$ because assuming disjoint charging points $M(t) = \cup_{i \in I} M_i(t)$ holds.

$$= \int_{-\infty}^{\infty} \underset{\sim}{l}(t) \, dt = \underset{\sim}{L} \tag{8}$$

□

According to the above proven theorem, the sharing of a number of CPs can be beneficial in a given infrastructure. The actual efficiency gains and related cost-savings per participant over time can differ depending on the actual values of $M$, $N$, $I$ as well as $M_i$, $N_i$ for each pooling participant, respectively. Despite the potential for higher usage, it can be assumed that tenants/users of a property will initially try to use exclusively their own self-installed charging points. They will consider sharing as an option, when bottlenecks of the exclusive usage of charging points become noticeable, e.g., frequently reported by frustrated EV drivers. A privileged EV-driver (e.g., a CEO of a company), who can exclusively use a particular parking space and the associated charging station, would not prefer to have that charging station included in the shared pool. However, in exceptional cases, e.g., due to failures, or maintenance work, such users could also benefit from the still functioning CPs of other entities. To ensure that some of the own charging points are always available for own purposes, a CP owner can decide to share only a subset of his charging points (see also the scenario in Figure 1).

## 5. Technical Feasibility of the Concept

The creation of a CP pool within an already existing, up and running EV-charging environment requires specific technical and non-technical tasks that have to be performed by each contributor. Figure 2 gives an overview of those tasks.

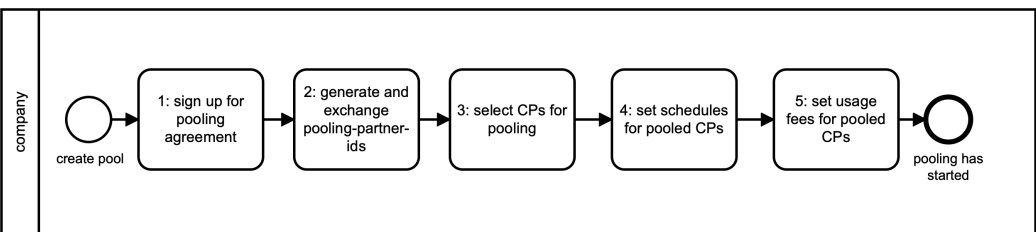

**Figure 2.** Main steps to initiate a pool of charging points.

As a first step (1), the owners of charging points can jointly negotiate and sign up for an agreement, e.g., in the form of a formal contract. The agreement can state how many or which charging points of a particular owner shall be added to the pool. It can also detail the usage conditions of pooled CPs and the respective parking lots. Examples are time-related aspects (validity period of the agreement, time intervals in which CPs are excluded from pooling), fees for using a pooled CP and the corresponding parking lot, payment modalities, and countermeasures in case of technical problems. In addition, the agreement can regulate operational aspects of the CP pool. For instance, the removal of CPs from the pool must be confirmed by each participant, or temporary access restrictions due to repair or maintenance work must be announced by the given CP owner in advance.

The operative management of pooled CPs can be carried out in a centralized or decentralized way. In a centralized setting, the agreement can explicitly name a dedicated entity (e.g., one of the pooling participants, or a contracted facility manager), who will take over responsibility for managing the CP pool. The managing entity processes all technical change requests concerning the pooled CPs and also helps onboard additional CP owners.

In a rather decentralized approach, the pooling-partners act more autonomously, retain administrative control and self-manage their own charging points. Consequently, they implement and run the CP pool in collaboration. Centralized pooling can reduce the efforts of individual CP owners by bundling all related tasks at one entity. Participating businesses must, of course, cover related costs and make sure that the managing party gets access to all necessary information and related systems. In multi-site environments (of larger companies), it is assumable that the CP pool at each site would be under control of a different (local) entity specific for that site, which can increase the overall complexity.

The main advantages of a decentralized management are increased flexibility and lower operating costs for the pooling participants. It enables each partner to control and

monitor its own charging points without further interaction or negotiations with some central party. A continued self-management of processes and related data might also ease the handling of multi-site scenarios. This approach also fits better with the above-mentioned assumption that CP pooling is more of an evolutionary step for companies, which they will take after it has become apparent that their installed capacities are no longer sufficient to meet their demand.

To initially create the pool, in the above-mentioned decentralized way, each participant can generate a unique secret and share it with the pooling partners over a proper trusted channel (see step 2 in Figure 2). There are many possible ways to implement this step. As an example, a CP owner can generate a random number and store it in the form of a QR code, which can be scanned by one or more respective pooling partners during face-to-face meetings via smart phone. The secret code received from a CP owner empowers the receiving pooling partners to co-use the pooled charging points (and only those) of that particular CP owner. For instance, the code can be used as an API key to get access to relevant data in the IT system in a controlled way. In the next step, each pooling partner selects and configures the (subset of its) charging points that shall be part of the CP pool (3), determines associated schedules and time restrictions (4), in case these were agreed on. For example, a particular pooled charging point might not be co-used on weekdays between 8:00 a.m. and 7:00 p.m. Afterwards, the monetary aspects of sharing, i.e., usage fees for each pooled charging point, can be configured (5). The fee for the co-usage could contain a fix component to cover the costs of general operations (maintenance and repair), and a variable part to compensate actual expenses related to EV charging, including cost of electricity, parking fee, taxes, etc.

Following the Business Process Model and Notation (BPMN) standard, the diagram in Figure 3 shows how an EV driver can use a pooled charging point within a semi-public infrastructure. An EV driver, say a restaurant guest, who has never visited the restaurant before, wants to use a charging point close to the restaurant entry. As the restaurant participates in the pooling initiative (refer to Figure 1), the guest can reserve any of the pooled CPs (incl. the parking slot). The reservation can be implemented as part of the usual table reservation procedure, in which the guest's credit card number is also captured. Note that the guest does not need to know, whom the selected charging point actually belongs to. Upon arrival, the guest parks her EV and identifies herself by presenting her credit card at the reserved charging point. In accordance with the available capacity, the local system triggers power scheduling for the particular CP and the charging starts. As the guest actually uses another tenant's charging point, the restaurant receives a notification. After the dinner, the guest accepts the restaurant's friendly offer to take over the costs for using the parking slot and charging the car's battery, so that she only has to pay for the expensive dinner. As a consequence, the restaurant will have to pay the calculated pooling fee to the CP owner as it was stated in the sharing agreement and also configured in the system. Note that, in case the guest would have used the restaurant's own charging point, these costs could be significantly lower (mainly electricity). Thereafter, notifications can be sent to all three involved entities about the related expense claims. Normally, the guest would drive away, i.e., free up the parking slot and the charging point, within a few minutes. As a result of the cost transfer, there will be no additional fees charged on her credit card. Instead, the CP owner will pay the actual electricity costs for the charging (as part of his usual electricity bill), and the restaurant's owner will pay the calculated sharing fee to the CP owner, as a compensation. The latter transaction can take place immediately after the charging process ended. Alternatively, the system may also collect such costs and conduct a periodic clearing between pooling participants. Should the host not agree to take over the costs for its guest's EV charging, the guest would have to pay the respective sharing fee to the actual CP owner, in the example case via a credit card payment transaction. The basic possibility for rejecting the transfer of costs can be helpful to prevent misuse of the charging infrastructure. If the guest has not finished EV charging or blocks the parking lot longer than a certain period of time, the restaurant, as host, may receive a warning about

the (meanwhile potentially much higher) sharing fee, so that it can refuse its generous offer. In this case, the EV driver can get a notification message and must pay the bill for the prolonged battery charging. In order to implement such measures against "unfriendly" drivers, additional proximity sensors at the charging points or a camera-based observation of the parking lots may be required.

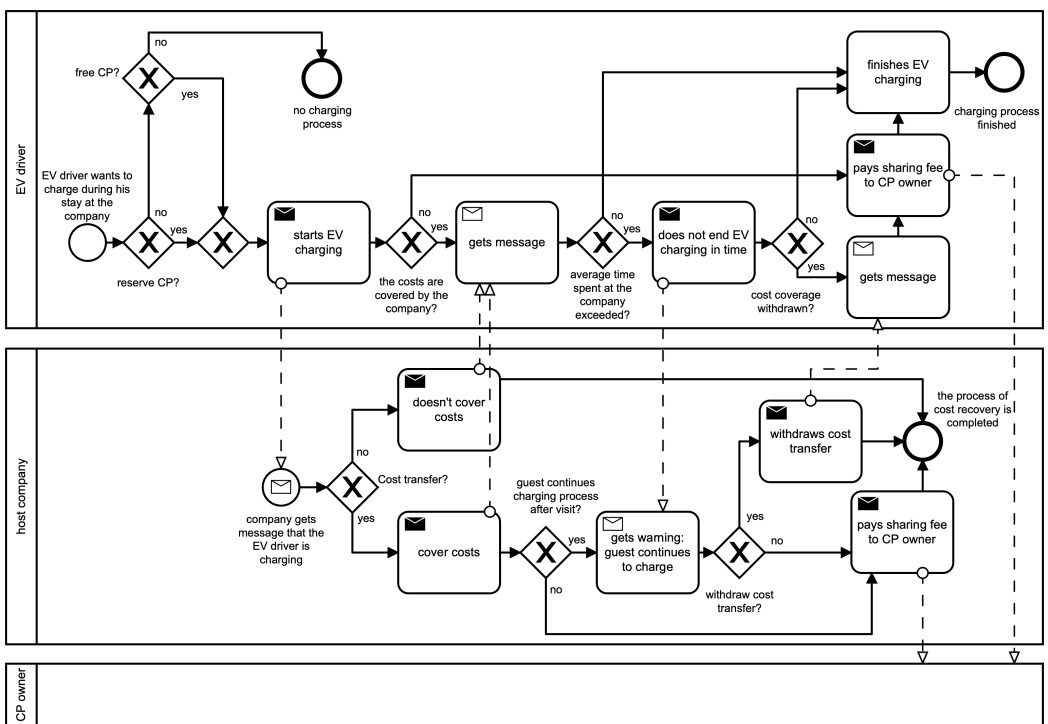

**Figure 3.** Example process: An external guest EV driver uses a pooled charging point.

The above described process can be significantly simpler, in case the EV driver is a known (permanent) user of the charging infrastructure and is assigned to a particular CP owner, e.g., her employer. The EV driver in this case can possess a personalized ID card. The card can be used for authentication at the company's own charging points and at every other pooled CP. The routing of related costs for "charge at work" to the employer's account can occur without any further interactions and notifications in a fully automated manner.

### 5.1. System Requirements

In order to run a pool of charging points in semi-public EV charging infrastructures, the corresponding software system must fulfil specific requirements, which we outline here (without claiming completeness):

- **Multi-tenancy:** Provide means of maintaining the data of multiple independent CP owners in one system allowing them to effectively monitor and control their part of the physical infrastructure at a given location in a safe and isolated way.
- **Autonomous administration:** Support the self-management of data about both shared and exclusively used charging points in the context of a given location. Enable CP owners to add/remove charging points to/from the pool, maintain potential time-based restrictions, pricing related information and other optional parameters concerning the shared usage of charging points.
- **Secure access control:** Manage the authentication and authorization of different end-users in case they try to use pooled or non-pooled charging points on site, or when they remotely access the system via an app or web-based user interface. Each CP owner has basically under its control whether and whom he or she might provide access to the charging points, since the overall infrastructure is considered to be "semi-public". An administrator representing a CP owner in the system shall be able to maintain

data that are needed to securely authenticate the internally known, registered end-users, such as related EV-drivers and maintenance staff. The system should allow the assignment of these users to the charging points of the given organization/tenant and manage related permanent credentials, such as personalized RFID badges. In addition, support the CP owner to grant external, previously unknown "guest" users (visitors, customers) temporary access and respective credentials to use the charging points for a limited duration or limited number of charging processes. The CP owner can send some secret (e.g., one-time password) upfront or hand out physical tokens (e.g., RFID cards) to its guests once they arrive. If the charging point supports the technology, previously unknown users can also be identified by using well-established systems, such as credit cards (via the NFC interface). In case an EV driver wants to use a pooled CP that does not belong to his organization/tenant, the system must check the user's presented credential against the authentication data of all tenants that participate in the CP pooling before rejecting the request. Should the user possess a valid credential issued by one of the pooling partners (whether permanent or temporary), the access must be allowed.

- **Reservation:** Offer charging point reservation capability for known, i.e., frequent or permanent users, and also for previously unknown ("guest") users. EV drivers shall be able to select and reserve each available charging point within the pool in addition to the visited host's own CPs. Various ideas and options of implementing a reservation system can be found in [31–33]. In case of yet unknown users, reservation must be coupled with an initial registration procedure in conjunction with the above-mentioned authentication mechanisms.

- **Cost and payment management:** Enable CP owners to maintain fix or variable fees for the co-usage of charging points. At the end of a charging process, ensure that all data items that are needed to calculate relevant expenses are captured correctly, such as consumed power (to calculate electricity costs), duration of charging (to calculate parking fees), etc. Ensure the correct routing of resulting pre-calculated expenses towards the right entities that are involved in the charging process (EV driver, host, actual CP owner) based on predefined rules and configuration. Implement interfaces to respective (external) rating, billing, invoicing, payment and clearing systems to further process actual payments and deliver related information in a secure way.

- **Power management:** Support the optimal utilization of available power capacities within the entire infrastructure [34,35], for example, in conjunction with a local energy management system. The overall system must fulfil the charging demands of EVs, while considering local power limits, total capacity and current state of charge of EV batteries, intended departure times and other relevant parameters. The pooled usage of CPs at a given location should not negatively impact the (previously established) load management system. Once a charging process has started, the power supply for the given charging point has to be managed independently from its membership in the pool.

- **Exception handling:** Implement measures to detect and handle exceptions in case of technical problems (e.g., when a pooled CP is out of order). Implement proper workflows and notifications to deal with the intentional misuse of pooled chargers, such as "long term parking".

### 5.2. Technical Details

In the context of a facility that hosts and serves multiple businesses, the facility owner may provide tenants with access to an integrated building management system (BMS). Such systems are broadly used today, for example, to control employees' access to floors and offices, book meeting rooms and co-working spaces, report problems, etc. BMS often allow the monitoring of energy flows in the entire facility. Procedures related to EV charging, including the option to self-manage and share own charging points with other co-users of the facility, could be implemented as integrated part of an overarching BMS. Another basic

option, which is more common practice today, is to run a specialized system for charging infrastructure management and onboard facility tenants as soon as they install CPs.

To implement our CP-pooling concept and demonstrate its technical feasibility, the open-source system "Open E-Mobility" has been chosen as the technical foundation [30]. "Open E-Mobility" supports the management of EV charging equipment at multiple sites of organizations that operate EV fleets. It has been already deployed in several productive and trial environments. The system can connect to and work with charging stations (both AC and DC) of multiple vendors and provide the operator with real-time status information about ongoing charging processes. Thanks to an integrated smart charging capability, the system can monitor and adapt the use of charging points to optimize energy consumption, while protecting the local grid against overloading.

The software system is designed to be deployed as a scalable cloud application. The high-level architecture is shown on Figure 4. The internal business logic and processes are implemented by the backend server built in NodeJs. The data that are created and managed in the backend persist in the form of document collections in MongoDB. The datasets of multiple CP owners, who are also termed as "tenants" of the system (as this term is commonly used in cloud applications), are maintained in isolated collections. With the help of the front-end server and respective views of the web-based graphical user interface (GUI), each tenant/CP owner can model its EV-charging infrastructure at multiple sites by assigning charging points accordingly. User access to the system and the underlying data of single tenants is controlled by means of roles (e.g., "Admin" or "Basic") that can be assigned to users. Furthermore, a "SuperAdmin" can conduct overarching tasks, including the creation and removal of tenants within the system. The communication between connected charging stations and the backend server is based on the Open Charge Point Protocol (OCPP) using both HTTPS and secure WebSocket as transport options. Each CP owner can maintain information about its users helping to authenticate them at the charging points (e.g., via RFID badges issued by the CP owner) as well as in the web application (via password). In addition, a CP owner can monitor the status of ongoing charging sessions and collect comprehensive logging information about relevant events. End-users can also access the system via a mobile application, for example, to view available charging points at a site, to trigger charging, or to obtain notifications (in form of emails).

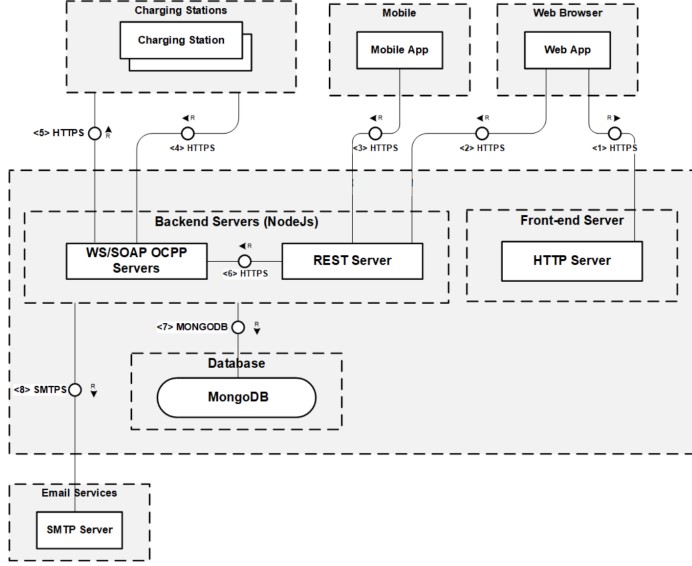

**Figure 4.** High-level system architecture of "Open E-Mobility". Reprinted with permission from [36]. 2022, SAP Labs France.

In the course of the prototype implementation, the publicly available source code of "Open E-Mobility" (see the current version on GitHub [36]) was extended, by adding

required functionalities to the server, creating new and altering existing database objects and by modifying user interfaces.

In order to maintain a pool of charging points, each CP owner manipulates its own local data within the boundaries of its own isolated tenant environment. With other words, the pool is not explicitly created as a system-wide entity that some super-user could maintain. Instead, all information about the pool is stored in the participating tenant's datasets. For that purpose, the previously existing user role "Admin" has been extended by additional activities related to CP pooling that a given CP owner has to perform. Accordingly, a user assigned to this role is allowed to generate a pooling ID (specific per cloud tenant) which is encoded in the form of a QR code. The user can also add pooling partners by scanning QR codes that he received from another CP owners/tenants. In addition, the user can mark charging points as "pooled" and also remove them from the pool. The user is also capable of manually approving (or deny) a requested transfer of costs, if a guest used a pooled charging point. The user role "Basic" has not been enhanced by new software capability related to CP pooling. However, a user in this role (typically an EV driver) will see in her app all charging points of the default-assigned CP owner (employer, business partner, host). In addition, she can also see all other pooled CPs as basically usable charging points. This is achieved by checking the database to see which tenants have actually joined the pool and which of their charging points are flagged as shared. To achieve these steps efficiently, a new document collection is created in each tenant's database in which data about the pooling partners of the respective cloud tenant are stored. With the help of this information, the server can quickly determine which tenants' datasets it should inspect. In case the tenant has no pooling partners, the collection would be empty. In the collection that stores data about the charging stations of the given tenant, the Boolean property "pooled" is added to the documents.

In order to start the charging process at a particular charging point, the authentication request sent by the CP is first checked against the user data of the respective CP owner. In case the user/credential is unknown or not valid, the authentication would normally fail and the charging point would reject to start charging the EV. Following our pooling concept, if the user/credential is unknown to the directly targeted CP owner, the server would check against (potentially all) other tenants' respective data collections whether the presented credential is known and valid and notify the requesting charging point about the result finally. During this procedure, the server uses the above-mentioned mechanism to select and inspect candidate tenant's data. In the current prototype version, all users of the cloud tenant receive access to all pooled charging stations of each pooling partner equally. The enforcement of potential restrictions, such as exclusion time intervals, is not yet supported. For that purpose, further properties could be added to the pooling partner collection scheme.

Concerning the graphical user interface, a new view for the administration of pooling partners was added and assigned to the user role "Admin". The view contains options for adding and removing pooling partners and the generation of a QR code to be exchanged with potential pooling partners. In addition, the already existing detail view of charging points is enhanced by a check-box to add a given charging point to the pool.

In the example screenshot shown in Figure 5, both plugs (termed as "*Connector A*", "*Connector B*") of the same charging station ("*CS-ABB-00001*") are added to the pool called "*Area1*". To explicitly allow or deny the cost coverage for unregistered guests and to carry out related monetary transactions once the charging has finished, a switch button "*Pay for guest*" as shown in Figure 6 was added to the detail view about ongoing charging sessions.

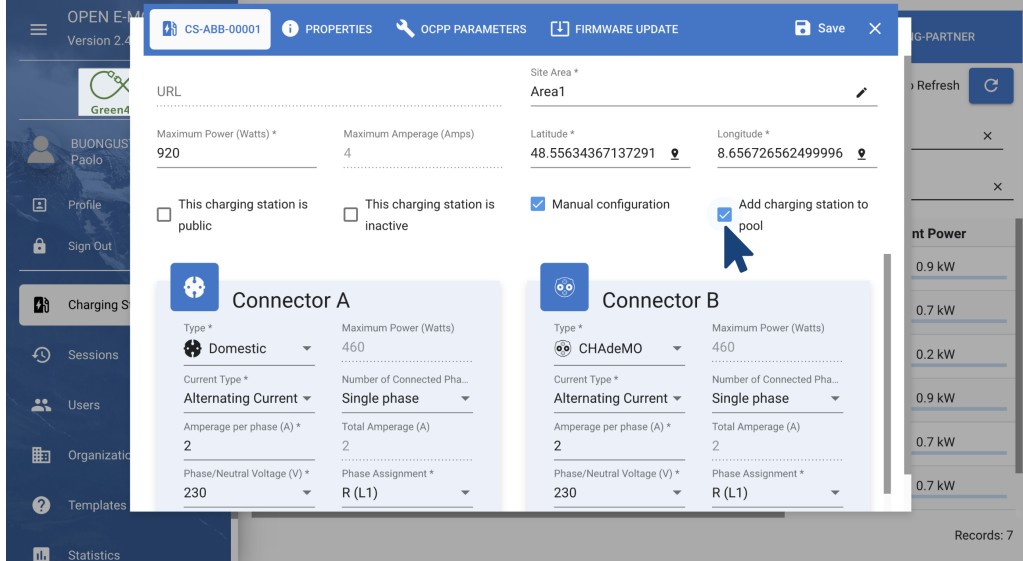

**Figure 5.** Example screenshot for adding charging points to an existing pool.

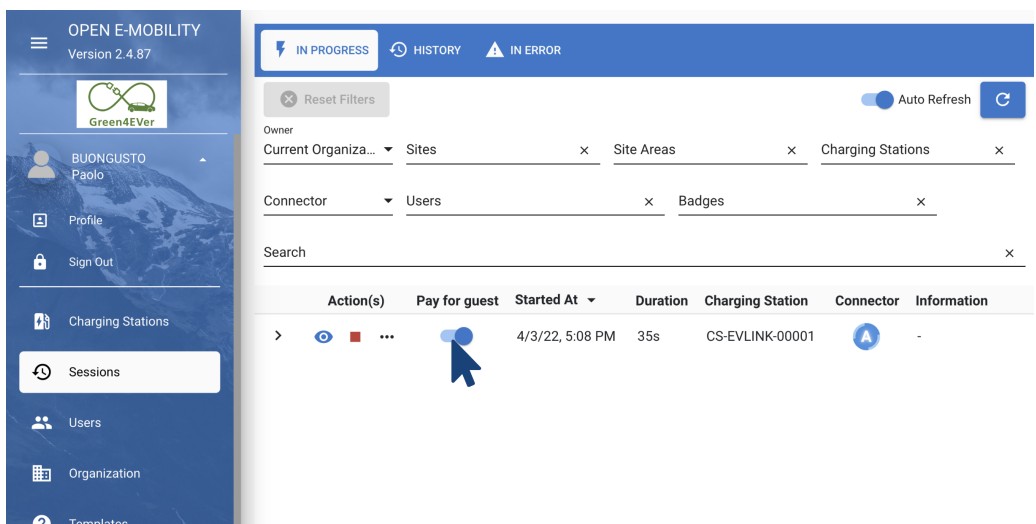

**Figure 6.** Example screenshot to initiate the transfer of costs of charging in the system.

The switch is visible for the end user together with other information about a running session (see "*IN PROGRESS*" in Figure 6). During the EV charging process, i.e., before the session terminates, an authorized user can turn on the switch. In that case, the backend server can set the respective tenant's account data to which related costs can be invoiced later. In an example, a restaurant owner could use the switch to cover the costs of EV charging, while guests are enjoying their dinner.

Note that the current implementation is not yet capable of preparing and launching actual monetary transactions with payment systems that are otherwise supported by "Open E-Mobility".

## 6. Desirability Survey with Potential End-Users

In order to capture and understand the preferences of potential end-users with regard to the sharing of charging stations, we reached out to companies in a small German town. In total, 46 companies were actively involved, including retailers (~30%), service providers (~24%), manufacturers (~10%), and others. They responded to an online questionnaire, which contained closed-ended questions asking the respondents to choose from a list of options. In addition to the online survey, we have also conducted personal interviews with

four selected retailers and with a restaurant owner. In the following, we summarize the results based on the questionnaire as well as the most important findings of the interviews.

### 6.1. Results of the Online Survey

One of the main objectives of the survey was to find out under what economic and technical conditions companies would share their existing or yet to be installed charging points. The corresponding questions, response options and the number of respondents, who have chosen the respective option, are shown in Table 1.

**Table 1.** Summary of company preferences.

| Question | Options | # |
|---|---|---|
| Would you pay the cost of charging for external guests? | Yes | 1 |
| | No | 15 |
| | Maybe | 27 |
| | No response | 3 |
| Whom would you offer EV charging for free? | Business partners | 10 |
| | Customers | 19 |
| | Employees | 22 |
| | Suppliers | 1 |
| How should EV drivers interact with pooled CPs? | Guest-card | 6 |
| | QR-Code | 11 |
| | E-roaming | 2 |
| | No preference | 27 |
| Which means of reservation would you prefer? | Time based | 8 |
| | 24h privilege | 5 |
| | Equal rights | 13 |
| | No preference | 20 |

Considering external guest users, e.g., the customers/visitors of other companies, only one respondent would cover the costs of EV charging (electricity, parking fees, taxes) without further restrictions. One-third of the survey respondents would definitely refuse to pay for external users' consumption.

The willingness to allow the usage of a company's CPs for free is significantly higher, if the respective EV driver has a business relationship to the company (business partner, customer, employee). However, only one respondent considered this option specifically for suppliers.

With regard to user interaction with the pooled CPs, each indicated response option (guest card, QR code, E-roaming) was illustrated and explained by a short process description, incl. the handling of related payments. As shown in Table 1, the majority (27) of respondents have not expressed any preference. The most popular option is to scan a QR code at the charging point in conjunction with a web application that manages further steps. Allowing the usage of roaming platforms (e.g., Gireve or Hubject) via the ID cards of supported e-mobility service providers was preferred by two companies.

Six of the respondents would prefer to hand out guest cards to authenticate their customers at the accessible charging points before the charging process would start.

The results also show that most survey participants (20) had no preference, whether and how EV drivers can reserve pooled charging points.

A relatively high number (13) were in favor of allowing CP reservations for all types of users equally. The scenario termed as "timed-based reservation", which received eight votes, allows CP owners to restrict the reservation and usage of their pooled CPs by others

to specific time intervals. Five respondents would keep exclusive reservation rights of their own charging points up to 24 h prior to the intended usage.

Survey results indicate that the shared usage of charging points is basically an interesting application for the involved companies. As survey participants do not prefer a particular technology or method, the charge point management system should offer multiple options to provide CP sharing functionality.

Note that the survey was conducted with a limited number of participants. They represent the business domain in a relatively small German town, where only a few charging points are in operation. Therefore, the responses and opinions should not be considered representative.

*6.2. Personal Interviews with Business Owners*

To further evaluate the developed concept of pooling charging points with other companies and to gather feedback on the prototype implementation, we carried out qualitative face-to-face interviews with five selected business owners. Thereby, we explained them the basic idea using a simplified version of the example scenario shown in Figure 1 comprising only two tenants (a restaurant and a physician). Note that these example businesses normally have less overlapping opening hours, making CP pooling especially advantageous for them. To gain comparable results from the interviews and reduce bias, a common presentation was used. Nevertheless, the slightly deviating wording of the questions in the individual interview sessions may have influenced the answers given by the respondents. As part of the presentation, we created a pool of CPs by using the software system and asked participants to experiment with the user interface during the interview sessions.

Overall, the concept of sharing charging points is considered useful by the interviewed business owners. In their view, it might be especially feasible for business owners, who know and trust each other, and have common customers. However, two of the interviewees commented that it was not easy for them to empathize with the challenges associated with EV charging because they do not yet own/use EVs themselves. The owner of a restaurant stated the relevance of the actual business location as potential obstacle: His restaurant is located outside of the town, and there are no other companies as potential pooling partners within walking distance. This feedback underpins that CP pooling is in general more beneficial in dense urban areas, where potential pooling partners are in close proximity to each other and the number of charging points per company is rather limited. Nevertheless, the restaurant owner is already looking for a practical solution to make his CPs available for selected guests as an additional service.

Since there are currently only a few charging points operated in town, one of the retailers suggested to rather establish charging points at the publicly accessible parking areas in the city center. In his opinion, it would be more meaningful than equipping the short-term parking spots at the companies' locations. Another retail manager would install charging points as prestige objects for his business and make the CPs exclusively available for his customers and employees. In contrast to this strategy, another interviewee values sharing as a "real bonus" for his customers.

These relatively diverse attitudes and approaches to the shared usage of charging stations might converge in the future. We assume that companies will demand additional charging points (especially at peak times during their specific business hours), if they cannot expand their own infrastructure anymore due to local limitations. Sharing CPs can be highly beneficial for companies with different business/opening hours. It was also confirmed by an owner of a retail store, who already shares expensive parking spots (currently without installed EV chargers) with a nearby restaurant.

The feedback given on the user-friendliness of the prototype system was overall positive. The interviewed business owners particularly highlighted the importance of a simple, user-friendly design and the support of mobile devices (especially for EV-driving employees). They suggested minor improvements concerning the GUI, such as the simple

selection of multiple charging points when the CP pool is initialized (instead of adding each CP separately).

Regarding the practical usability of CP pools, interview participants asked several questions that are very valuable for improving the system. Two example questions were: How could a first-time customer know whether or not the cost of charging would be covered by the respective business? How could such customers know which charging points are already reserved? These communication-related problems can be solved, if EV drivers are willing to install and use a specific mobile application that visualizes the status of CPs (e.g., available, occupied, reserved) within the local charging infrastructure. Most business owners consider the usage of such a dedicated app for their customers (with their words) "annoying". In case a business refuses the usage of a provided mobile app, the EV-drivers could receive static information only. For example, signs, stickers, or printed instructions placed at the charging points could indicate whether a CP is part of a pool. Only one interview partner assumes that EV-drivers would be willing to install a specific app for this purpose. One business owner suggests to cooperate with an e-roaming provider that already enables its registered EV drivers to search for and reserve available CPs (at publicly accessible parking/charging locations).

## 7. Conclusions and Future Work

In this paper, we described main results of the conceptual phase of our ongoing research project. We presented the basic idea to establish and run shared EV-charging infrastructures in the exemplary context of commercial real-estate facilities. Collaborating EV-charger owners, typically companies that are using the same facility (office building, industrial area), jointly create, manage and use a pool of EV chargers. A pooled charger can basically be used by all employees, business partners, customers, guests, etc. of the pooling participants. In our work, we have been considering aspects of economic viability, technical feasibility and desirability in order to create a successful solution.

The economic viability of the solution is of interest for both the solution vendor and the companies that use it. Our related investigation focused more on the users' potential economic benefits. We have shown that CP sharing can help improve the overall utilization of a particular charging infrastructure. As a consequence of a higher utilization, involved companies could reduce investments in EV charging equipment because the chargers of others can substitute self-installed chargers to a certain degree. It is similar for operational costs (inspections, maintenance and repair work) that are in general proportional to the number of installed chargers. From the viewpoint of individual EV drivers, the possibility to use a (larger) shared pool of chargers will increase the likelihood to find an available charging point, e.g., when arriving at work. This is especially to the benefit of the "average" driver, who has no dedicated parking slot with an exclusively accessible charger. To ensure that some of the own charging points are always available for business-critical purposes, e.g., to charge delivery vans, service cars or shuttles, a company can decide to share only a subset of its installed charging points. Applying additional, e.g., temporal restrictions for the shared CPs can also help avoid unwanted situations. The proper selection and optimization of these parameters is a complicated task for companies. It is difficult to predict the demand for available charging stations at a given time, as it depends on the sometimes volatile behavior of their own and external users. We plan to address this problem as part of our future work by running simulations based on random generated and, if available, real datasets, incl. EV departure and arrival times, batteries' state of charge, etc. In addition to improving the infrastructure's utilization, the sharing of CPs can help cover the increased demand for charging capacity caused by exceptional events, such as a wedding party in a restaurant that does not own a sufficient number of chargers to serve all guests.

Despite the above-mentioned advantages, further efforts must be made to validate the profitability and feasibility of CP pooling. It can be yet assumed that in the corporate context the purely economic interest in sharing charging points with other companies

would be a less important motivating factor. However, improving customer satisfaction (through the high availability of charging points) can lead to a measurable positive financial impact in specific business areas, such as gastronomy, retail and hotels. National and international regulations, laws and taxation rules can heavily influence whether real-estate investment and management corporations—as building and facility owners—would invest into e-mobility equipment in future. Our current findings show that, at least in Germany, they mostly install no or only a few charging points in new buildings and related parking areas, but allow users/tenants to build and operate own equipment if needed. As of today, it is therefore less likely that a building or facility owner would provide its tenants with access to a software system to manage their own charging points. Without such a commonly used system, however, the creation and operation of a shared pool of CPs can be a challenging task.

In order to explore and understand potential users' preferences, we conducted an online survey with 46 company representatives in a German town. Five of them have also participated in in-depth personal interviews. Results of the survey showed that companies have different opinions and diverse requirements regarding the shared use of EV chargers in future. A technical solution should therefore be as flexible as possible to meet individual needs. Accordingly, we specified and modelled important processes, and formulated technical requirements that a management system should fulfil. The technical feasibility of the concepts has been successfully proven through the development of a software prototype based on the open source system "Open E-Mobility". The required modifications and extensions, especially regarding elements of the existing user interface, led to affordable development efforts. The system was tested with regard to the functional correctness of the software and the user interface. It has also been used in the above-mentioned face-to-face interviews and received positive feedback from the interviewees.

The prototype was so far only used to validate the concept and gather qualitative feedback of potential users for evaluation purposes. As of today, our approach to share charging points between independent business entities has not yet been tested in a real, productive environment. The current lack of quantitative test results gathered in a real-world setup motivates further activities. Together with our project partners, we are preparing a field test in the context of a large office building under realistic conditions. It is planned to actively involve the employees of several companies that are co-using the facility. EV drivers will thereby jointly use a "sandbox" charging infrastructure on site. The tests will run for several months helping to observe usage patterns and to document related energy consumption, infrastructure efficiency, costs and other relevant factors.

**Author Contributions:** Conceptualization, Z.N.; methodology, J.G., S.G., Z.N.; software, J.G.; validation, J.G., S.G.; formal analysis, J.G., Z.N.; investigation, J.G., S.G.; resources, J.G., S.G.; data curation, J.G., S.G.; writing—original draft preparation, J.G., S.G., Z.N.; writing—review and editing, Z.N.; visualization, J.G., S.G.; supervision, Z.N.; project administration, Z.N.; funding acquisition, Z.N. All authors have read and agreed to the published version of the manuscript.

**Funding:** As part of the research project "Green4EVer" [37] this work has been funded by the German Federal Ministry for Economic Affairs and Climate Action (BMWK).

**Institutional Review Board Statement:** Not applicable.

**Informed Consent Statement:** Informed consent was obtained from all subjects involved in the study.

**Data Availability Statement:** Not applicable.

**Acknowledgments:** We especially thank our project partners and colleagues at Art-Invest Real Estate, Webasto, and SAP for their kind support and expertise provided. In addition, we express our gratitude to Timo Kuch for his assistance with the mathematical formulations.

**Conflicts of Interest:** The authors declare no conflict of interest.

## Abbreviations

The following abbreviations are used in this manuscript:

| | |
|---|---|
| API | Application Programming Interface |
| BEV | Battery Electric Vehicle |
| BMS | Building Management System |
| BPMN | Business Process Model and Notation |
| CP | Charging Point |
| EV | Electric Vehicle |
| GUI | Graphical User Interface |
| HTTPS | Hypertext Transfer Protocol Secure |
| NFC | Near Field Communication |
| OCPP | Open Charge Point Protocol |
| PHEV | Plug-in Hybrid Electric Vehicle |
| RFID | Radio Frequency Identification |
| TCO | Total Cost of Ownership |

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
