# Peer review of "On the Collaborative Use of EV Charging Infrastructures in the Context of Commercial Real Estate"

_wevj, doi:10.3390/wevj13120223_

Round 1

Reviewer 1 Report

The authors presented a novel approach to pool EV-charging points in order to enhance the exploitation of these infrastructures in the context of commercial real-estate facilities used by multiple tenants. The topic is very interesting, but the article needs major updates before publication:

1) Improve the English style;

2) In the reviewer's opinion, the Related Work section should follow the Introduction section. In so doing, it will be possible to better appreciate the difference w.r.t. existing solutions;

3) Proof of Theorem 1 should be rewritten in a different way. More in detail, it will be better to describe it step-by-step (insert the description of each step after each equation);

4) A table that summarizes results could improve the clarity and readability;

5) Improve the Introduction and Related Works sections. For instance, the following could be added ([D’Acierno, L., Tanzilli, M., Tescione, C., Pariota, L., Di Costanzo, L., Chiaradonna, S., & Botte, M. (2022). Adoption of Micro-Mobility Solutions for Improving Environmental Sustainability: Comparison among Transportation Systems in Urban Contexts. Sustainability14(13), 7960.], [Caiazzo, B., Coppola, A., Petrillo, A., & Santini, S. (2021). Distributed nonlinear model predictive control for connected autonomous electric vehicles platoon with distance-dependent air drag formulation. Energies14(16), 5122.], [Lee, J. H., Chakraborty, D., Hardman, S. J., & Tal, G. (2020). Exploring electric vehicle charging patterns: Mixed usage of charging infrastructure. Transportation Research Part D: Transport and Environment79, 102249.], [Huang, X., Lin, Y., Lim, M. K., Zhou, F., Ding, R., & Zhang, Z. (2022). Evolutionary dynamics of promoting electric vehicle-charging infrastructure based on public–private partnership cooperation. Energy239, 122281.], [Simolin, T., Rauma, K., Viri, R., Mäkinen, J., Rautiainen, A., & Järventausta, P. (2021). Charging powers of the electric vehicle fleet: Evolution and implications at commercial charging sites. Applied Energy303, 117651.]).

Author Response

Dear Sir or Madam, thank you very much for your feedback and suggestions to improve our paper. 

Regarding your comment 1) below: Could you please point to sentences, which were difficult to understand, or make it otherwise clearer for us what improvements you meant?  

2) is done, related work became now section 2. 

3) Single steps of the proof are now commented/explained as mentioned. 

4) We assume you meant a table that summarizes the Evaluation results in chapter 4, is that correct? In the original conference paper, there was no room for such tables, etc. due to the 12-page-limit. Please let us know. 

5) We could download almost all papers you suggested and will reference them within intro and/or related work, as appropriate, thank you.   

Reviewer 2 Report

This paper covers an interesting and important area of study and there is potential for useful output. However, I don't feel it is suitable for journal publication currently.

Much of the content is a hypothetical discussion of how things may work rather than a description of the actual work that has been performed in the study. It is not until section 3 that the reader discovers that the actual research performed was a survey. If the results of this are the key findings then the process undertaken should have been described in a methodology section and the results clearly described and conclusions drawn. Rather the paper reads like a discussion of the area rather than a scientific article.

The related work would have been better discussed earlier in the paper to help identify the gap in knowledge and the research question, a clear methodology to address this question is also not clearly articulated. 

I also felt that the proof didn't belong in, or add anything to, the paper as it was just showing that if all charge points were pooled then there would be greater utilisation than if none were, which is an obvious conclusion.

Author Response

Dear Madam or Sir, 

thank you very much for your review and comments. Based on them as well as on the feedback given by the other reviewers I have updated the document. 

Regarding your point "Much of the content is a hypothetical discussion of how things may work rather than a description of the actual work that has been performed in the study. It is not until section 3 that the reader discovers that the actual research performed was a survey. If the results of this are the key findings then the process undertaken should have been described in a methodology section ..." I have included a new section "Methodology", see Section 2.

I moved and slightly extended the section on related work by referencing more sources (as pointed out by another reviewer). Now it became Section 3.

In my view it is an interesting finding that we can formally describe resource sharing in a pool (I have not seen similar yet) and then definitely show (not just "feel") that it can be of advantage for the participating resource owners.  Of course, you may see it different... 

Reviewer 3 Report

The paper addresses an interesting and complex problem of sharing EV charging infrastructure. It is complex since it consider design and implementation aspects such as scenarios, system architecture, configuration (contracts), usability, and preliminary evaluation. Very good points about configuration management end user guidance and usability have been provided by the feedback obtained from interviews and questionnaires.

Some scenarios are more beneficial than others. For example a short visit  to a restaurant or  a physician would provide a parking lot, but add little to the battery state of charge. Sharing outside the working hours would benefit residents in the neighborhood.

However, the mere analysis and design of the system aspects makes the paper an important contribution.

Please clarify  Figure 5 : what is the selected pool, do the two connectors A and B belong to the the same CP? In Fig 6:  when is the transfer of costs initiated? During the charging? Who performs it?

Author Response

Thank you very much for your feedback. Meanwhile I have changed/improved the clarifications regarding both Fig 5 and Fig 6. I will attach the resulting file, once we incorporated the comments of the other reviewers too.  

Round 2

Reviewer 1 Report

The authors improved the quality of the paper.  However, major improvements are still needed:

1) improving English style means using more formal sentences throughout the paper. For instance, the first sentence of Paragraph 4.1 could be rewritten as "Despite the above-mentioned potential for higher usage, it can be assumed that tenants/users of a property will initially try to use exclusively their own self-installed charging points". 

2) The reviewer means a Table that summarizes the Evaluation Results in Chapter 5. 

3) It would be better to insert all the articles suggested in the last review round. 

Author Response

Thank you very much for your recent comments, highly appreciated.

In the new version of the paper, we have

added a Table (see Section 6.1) and explanations concerning the most important questions and related distribution of the answers given by questionnaire participants 

included ALL papers you have suggested (see References and Section 2 on Related work) 

tried to further improve En style overall ...

Best regards

Reviewer 2 Report

The addition of a methodology section helps the paper but needs more work as does the overall structure of the paper in my view.

* The related work section should come after the introduction and clearly show and articulate the knowledge gap that is being addressed in the paper. (The section numbers also don't match the sections in the paper currently)

* The methodology needs more work and clarity about what has been done to target the 3 areas of focus that have been highlighted. More details are required e.g. how many businesses? What kinds of businesses? Are they representative of a diverse pool of sharers? What did the questionnaires cover? How were they analysed? How will the software system be tested and validated specifically? ...

* I think the following sections should be structured as Results for these 3 areas i.e. commercial viability (current section 4), desirability ( current section 5), technical feasibility (sections 4.1 to 4.3)

* Conclusions are required to state what has been done and shown for the 3 aspects of the methodology. The abstract should also summarise the methodology, key findings, and conclusions.

Author Response

Dear Madam or Sir, 

thank you for your below comments and suggestions to improve the paper. 

In the new re-edited version you will find the following changes following up on your hints:

  • Related work became Section 2 as suggested. The section numbers at the end of the intro fit now to the new structure of the paper that has now 7 sections.
  • In section 3 have added more information about our methodical approach to target the three dimensions of interest (viability, desirability, feasibility). 
  • The results wrt these dimensions are now described in separate Sections as you suggested. Section 4: Viability, Section 5: Technical process models and details of the Proof-of-concept implementation Section 6: Questionnaires and interviews. 
  • Section 7 concludes the results accordingly and I described our future work plans in more detail. 

Reviewer 3 Report

The scarcity of resources (parking places and charging power) are the main reasons for two owners  to pool some of their charging points. What if the service after pooling becomes worse for some users? It should be mentioned how parameters for pooling can be set and adjusted and which data is used for that.

Author Response

Dear Madam or Sir,

thank you very much for your below comments, 

In the new version of the paper I pointed out that especially highly privileged users (those with an own parking slot and charger) would less profit from and thus rather dislike the sharing idea. Therefore, it is a good strategy to share only a subset of own CPs. See related thoughts in Section 4 (following the formal proof). 

A good selection (optimization) of the parameters, like time intervals for sharing, which / how many CPs to pool, etc. is topic for future research as mentioned in Section 7 (outlook). We do not have yet a proper algorithm for it... 

Best regards

Round 3

Reviewer 1 Report

The authors addressed all the issues. The paper can be published in the current form.

Author Response

Thank you very much for your feedback. 

Reviewer 2 Report

The paper is much improved and is now largely well-structured. However,  I think the results need to be more clearly associated with the methodology. I would suggest naming the results sections the same as the 3 issues identified in the methodology (Economic viability, desirability, technical feasibility) and ensuring they are delivered in the same order as bulleted in the methodology.

Author Response

Thank you very much for your feedback. 

We renamed the respective sections (4, 5, and 6), and re-ordered the bulleted list  of dimensions in section 3 accordingly.  

Reviewer 3 Report

Empirically, the advantages of pooling the CPs can be understood, and making use of a partial implemented prototype in order to get feedback about the usability of the system is a legitime approach for presenting the concept.

However, a research paper is in my opinion not completed without an  quantitative system evaluation. Pooling decisions between partners may improve the charging throughput of the system, poor decisions or changes in the charging demand might for example decrease the service availability for one of the pooling partners. A model would provide the optimal pooling configuration and quantitative data based on the charging demand pattern, improving the score for research design considerably.

The paper could be accepted, if it clearly states the conceptual phase and the limitation to qualitative results and planning considerations.

Author Response

Dear reviewer,

thank you for your remarks concerning the need to continue our research work in order to gather quantitative results as well. It is of course a valid point and we have updated section 7 accordingly.  

Best regards
